# Neoadjuvant Endocrine Therapy in Breast Cancer Management: State of the Art

**DOI:** 10.3390/cancers13040902

**Published:** 2021-02-21

**Authors:** Florence Lerebours, Luc Cabel, Jean-Yves Pierga

**Affiliations:** 1Medical Oncology Department, Institut Curie, 92210 Saint-Cloud, France; luc.cabel@curie.fr (L.C.); jean-yves.pierga@curie.fr (J.-Y.P.); 2Department of Medicine, University of Paris, 75006 Paris, France

**Keywords:** breast cancer, neoadjuvant, endocrine therapy, prognosis

## Abstract

**Simple Summary:**

Over the last ten years, neoadjuvant endocrine therapy (NET) has been increasingly investigated and has gained recognition. NET should not only be used to allow surgery or to improve breast-conserving surgery rates in patients not eligible for NCT, but also as a research tool for the search for endocrine sensitivity biomarkers and targeted therapies, and for prognostic information in ER+/HER2-.

**Abstract:**

Endocrine therapy is the mainstay of treatment in HR+/HER2- breast cancers, which represent about 70% of all breast cancers. Neoadjuvant therapy has been developed since the 1990s to address several issues, including breast-conserving surgery (BCS) and improvement of survival rates. For a long time, neoadjuvant endocrine therapy (NET) was confined to frail patients in order to improve surgery outcome. Since the 2000s, NET now plays a central role as a research tool for predictive endocrine sensitivity biomarkers and targeted therapies. One of the major issues in early HR+/HER2- breast cancer is to identify patients in whom chemotherapy can be safely withheld. In vivo assessment of response to NET might be the best treatment strategy to address this issue.

## 1. Introduction

The primary goal of neoadjuvant systemic therapy in breast cancer is to downstage tumor size to allow breast-conserving surgery (BCS) or mastectomy when the tumor is initially inoperable, while the role of neoadjuvant therapy on axillary downstaging is still under investigation. The neoadjuvant setting also provides the opportunity to monitor response to treatment and can therefore help to identify predictive factors and biomarkers. Neoadjuvant therapy constitutes a unique opportunity to test new drugs and alternative treatments when primary systemic therapy is not sufficiently effective. Recent examples include treatment with trastuzumab emtansine in HER2-positive or capecitabine in triple-negative breast cancers in the presence of residual disease after neoadjuvant chemotherapy (NCT) [1,2]. As achievement of a pathological complete response (pCR) with NCT is correlated with outcome, pCR has become the gold standard to assess the efficacy of NCT and was approved by the FDA in 2014 as an acceptable surrogate endpoint of clinical benefit in clinical trials of neoadjuvant therapy for accelerated drug approval. Of note, the magnitude of the benefit of pCR varies according to the molecular subtype of the breast tumor [3]. In particular, HR-positive/HER2-negative breast cancers, which account for approximately 70% of all breast cancers, display low pCR rates (<10% for grade 1–2, ~15% for grade 3) [3]. Moreover, the survival gain due to this pCR is smaller than that observed in aggressive subtypes. It is generally accepted that these luminal tumors could derive a greater benefit from endocrine therapy (ET) with the advantage of being less toxic than chemotherapy. The treatment guidelines for HR+/HER2- metastatic cancer are a perfect illustration of this situation [4].

Until the 2000s, neoadjuvant endocrine therapy (NET) was mainly used for elderly and frail patients. Subsequently, prospective NET trials, including phase III trials conducted on younger postmenopausal HR+ women, provided evidence in favor of NET with clinical response and BCS rates similar to those reported with NCT. International guidelines consider that NET given for 4 to 8 months is a validated treatment in postmenopausal women with hormone receptor-positive HR+/HER2-negative (HER2-) breast cancer to improve surgical outcome and to allow BCS [5,6].

The purpose of this review is to discuss the practical modalities of NET, current research on predictive factors and new drugs, and finally to define the population to whom NET should be offered.

## 2. Endocrine Agents

The efficacy of NET was initially demonstrated with tamoxifen (a selective estrogen receptor modulator (SERM)) with an objective clinical response rate (complete or partial response according to RECIST criteria) of approximately 40% [7]. Importantly, it has also been shown that neoadjuvant tamoxifen followed by surgery gives the same overall survival rate as surgery followed by adjuvant tamoxifen in elderly women (≥70 years old) [8].

In the early 2000s, three phase III trials demonstrated the superiority of aromatase inhibitor (AI) (letrozole or anastrozole) over tamoxifen in terms of clinical response rate and breast conservation (Table 1) [9]. The primary endpoint in these three trials was clinical response rate based on palpation and/or ultrasound. The P024 trial compared 4 months of letrozole to tamoxifen before surgery. Clinical response and BCS rates were both significantly improved with letrozole [10]. The IMPACT trial was a 3-arm trial comparing anastrozole to tamoxifen or a combination of the two drugs. No significant difference was observed between these three arms in terms of clinical response or BCS [11]. The PROACT trial compared 3 months of anastrozole versus tamoxifen. The BCS rate was significantly improved with anastrozole, whereas the clinical response rate was numerically superior but not significantly different [12]. Based on these trials, aromatase inhibitors became the gold standard for NET in postmenopausal women. More recently, a meta-analysis of trials comparing neoadjuvant tamoxifen to AI confirmed the superiority of AI [13]. In 2011, a phase II trial, ACOSOG Z1031, did not find any superiority between the three AI: anastrozole, letrozole, and exemestane [14] (Table 1).

Fulvestrant is a selective ER degrader (SERD) that demonstrated a higher efficacy than AI as first-line therapy in the metastatic setting [15,16]. The NEWEST trial compared 16 weeks of neoadjuvant fulvestrant at doses of 250 to 500 mg. Tumor response rates at week 16 were similar between the two dosages (22.9 and 20.6%, respectively) [17]. We recently published a pooled analysis of two sister phase II trials evaluating anastrozole and fulvestrant as NET in postmenopausal HR+/HER2- breast cancer patients (CARMINA02 and HORGEN trials) [18,19,20]. These two trials had a similar design and included similar populations and the primary endpoint in both studies was clinical response rate at 6 months. CARMINA02 and HORGEN trials were non-comparative trials and were therefore not designed to assess the superiority of either of the endocrine treatments in terms of clinical response. We observed about 15% difference in clinical response rates in favor of anastrozole (55.9% in the anastrozole arm, 44.3% in the fulvestrant arm). No significant difference was observed between the two drugs in terms of BCS, pathological response, and survival rates. 

The phase III ALTERNATE trial randomized women with T2–4 N0-3 M0 ER+/HER2− invasive breast cancer to receive either anastrozole or fulvestrant or a combination of the two as NET with 434, 431, and 434 patients evaluable in each arm, respectively. Preliminary results showed that neither fulvestrant nor fulvestrant + anastrozole improved the rate of endocrine-sensitive disease compared to anastrozole alone. The rates of endocrine-sensitive disease (defined using pathological response to NET) were 22.7% vs. 20.5% vs. 18.6% in each arm, respectively [21].

A single phase III trial, the STAGE trial, evaluated the efficacy of NET in premenopausal patients in a population limited to 98 patients treated with anastrozole + goserelin versus 99 treated with tamoxifen + goserelin, with good results in terms of clinical response rate and breast conservation, and again confirming the superiority of AI (Table 1) [22]. In premenopausal patients, NET should therefore include an AI plus ovarian function suppression (OFS). In the GEICAM 2006-03 trial comparing NCT to exemestane (+ goserelin in premenopausal women), 51 patients (46%) were premenopausal. A significant benefit of CT over ET was observed in premenopausal women in terms of clinical response rate, with response rates of 75% and 44%, respectively (*p* = 0.027) [23]. This result of an unplanned exploratory analysis of this trial should be interpreted cautiously. However, in the absence of sufficient data, the use of neoadjuvant endocrine therapy should be reserved to postmenopausal women. One of the outstanding issues in this population is the potential lack of efficacy of chemical ovarian function suppression.

## 3. Comparison between NET and NCT

There are very limited data comparing NCT to NET, but the most informative data are derived from two phase II trials (Table 2). Semiglazov et al. randomized 239 postmenopausal women with stage II–III breast cancer (BC) to receive either anastrozole or exemestane for 3 months vs. NCT (4 cycles of doxorubicin plus paclitaxel) [24]. No statistically significant difference between NET and NCT was observed in terms of clinical (primary endpoint) and pathological response rates, but there was a trend in favor of NET for BCS rates. The abovementioned trial from the GEICAM 2006-03 randomized 97 BC patients to 6 months of exemestane (+/− goserelin) versus AC followed by docetaxel [23]. Except for clinical response rates in premenopausal women and those with high Ki67 expression levels, no statistically significant difference in terms of clinical response (CR), pCR, and BCS rates was observed between the two groups. Similar results for NET were reported in a phase III trial randomizing NCT vs. NET in premenopausal women with HR+/HER2- lymph-node positive BC [25]. A significantly higher number of patients obtained an objective CR or pCR with NCT compared to NET. No significant differences were observed in terms of BCS rates or Ki67 changes. The NEOCENT trial was a feasibility and translational study randomizing ER-rich BC patients to receive 6 FEC100 or letrozole for 18–23 weeks [26]. The trial was stopped after 44 patients had been randomized because of slow accrual. No significant difference was observed between NCT and NET in terms of radiological response rate. A meta-analysis of trials comparing NET with AI and NCT demonstrated no significant difference in terms of clinical response rate (OR: 1.08; 95%CI: 0.50–2.35; *p* = 0.85; *n* = 378), radiological response rate (OR: 1.38; 95%CI: 0.92–2.07; *p* = 0.12; *n* = 378), pCR (OR: 1.99; 95%CI: 0.62–6.39; *p* = 0.25; *n* = 378), or BCS rate (OR: 0.65; 95%CI: 0.41–1.03; *p* = 0.07; *n* = 334), but with higher toxicity with NCT [13]. Altogether, these data illustrate the poor risk–benefit ratio of NCT compared to NET in HR+/HER2- BC subtype.

## 4. Duration of NET

The duration of NCT ranges from 3.5 to 6 months depending on the number of cycles and/or dose-dense schedules. The optimal duration of NET remains controversial, as the time to best response is longer with endocrine therapy than with chemotherapy. As shown in Table 1, most NET trials administered NET for 3 to 5 months. Some studies have suggested the need for longer treatment. A median treatment duration of 7.5 months of letrozole was optimal for maximal tumor volume reduction and a BCS rate as high as 69% [27]. Another study comparing 4, 8, and 12 months of neoadjuvant letrozole demonstrated that 12 months of treatment resulted in higher clinical and pathological response rates [28]. Similarly, 6 months of neoadjuvant exemestane therapy reduced mean tumor size to a greater extent than 3 months of therapy in the TEAM IIA trial [29]. In the CARMINA02 trial, in which the duration of neoadjuvant therapy with anastrozole or fulvestrant ranged from 4 to 6 months, we also observed a consistent reduction in tumor diameter beyond 4 months of treatment [19]. In conclusion, in most studies, tumor response rates increased with treatment durations longer than 3 months [7]. In routine practice, NET should be proposed for at least 4 to 8 months [5,6]. The absence of clinical progression should be verified at 1 month and the response should then be assessed at 3 or 4 months. Development of secondary resistance to ET may be observed if ET is administered for more than 8 months [30]. Finally, it seems important to maintain a surgical project after completion of neoadjuvant therapy unless contraindicated by the patient’s condition.

## 5. Surgical Issues

Improvement in BCS rates was consistently reported in NET trials, more frequently with AI than with tamoxifen, and is one of the major goals of NET. Approximately 45% to 50% of patients who would require upfront mastectomy will be converted to BCS after NET [31]. However, surgical outcome as endpoint requires careful baseline evaluation of the radical surgery rate, and eligibility for BCS remains subjective. 

Although overall survival is similar after either neoadjuvant or adjuvant ET, higher local relapse rates were observed with NET than with primary surgery [8]. Similar results were reported for neoadjuvant versus adjuvant chemotherapy in the NSABP B18 trial as early as 1997, which were confirmed by a meta-analysis of NCT trials [32,33]. As for neoadjuvant tamoxifen, NCT was associated with more frequent local relapse than primary surgery followed by adjuvant chemotherapy, but overall survival did not differ regardless of primary treatment. However, this local relapse rate could be overestimated by inclusion in the meta-analysis of several clinical trials omitting surgery in patients with a complete clinical response to NCT [33,34,35,36]. The results of earlier studies must be interpreted cautiously, as staging, medical treatments, surgical techniques/margins, and radiation protocols should be considered to be inadequate by current standards. These data underscore the fact that surgery must nevertheless be maintained following NET.

Few studies have assessed the place of axillary lymph node surgery after NET and the few published results are derived from relatively old studies, at a time when initial axillary staging did not necessarily include ultrasound (US) +/− fine-needle aspiration or biopsy. It is noteworthy that patients treated by NET are less likely to undergo axillary lymph node dissection (ALND) than those receiving NCT [37]. Despite similar rates of axillary node involvement at diagnosis, patients treated by NET in the ACOSOG Z1031 B trial less frequently underwent ALND than those treated by NCT [14]. In a retrospective survival analysis of 4496 patients from the National Cancer Database who received NET for cT1–3N0–1M0 breast cancer between 2010 and 2016, Kantor et al. reported that survival of NET-treated patients mirrored that of upfront surgery patients [38], suggesting the possibility of considering de-escalation of axillary management strategies in NET patients. The ACOSOG Z1071 trial found a high false-negative rate (12.6%) for sentinel lymph node biopsy among women with breast cancer and initial axillary involvement (cN1) receiving NCT [39]. Clinical trials evaluating sentinel node biopsy in cN1 are ongoing, albeit for patients treated with NCT. However, the results of these trials cannot be directly extrapolated to axillary management after NET, as the prognostic value of residual nodal disease appears to be greater after NCT than after NET [38]. 

## 6. Assessing Response to NET

### 6.1. Clinical and Radiological Response

Until recently, the clinical response assessed by palpation with a caliper or by breast ultrasound and recorded using RECIST criteria was the primary endpoint most frequently used in NET trials. However, measurement by clinical palpation may be inaccurate, particularly in the lobular subtype or depending on fibrosis/edema that may occur during treatment. MRI has been shown to be more accurate than clinical palpation, US, and mammogram for measuring residual tumor size after NCT [40]. In the CARMINA02 trial, US at 1 month and before surgery was predictive of pathological response, while the contribution of MRI to response assessment was disappointing [19]. However, in a prospective study of 57 patients treated by NCT or NET, Takeda et al. reported a good correlation between residual tumor size as measured by MRI and pathological tumor size after both NCT and NET [41]. An ancillary study of the CARMINA 02 trial showed that early metabolic response assessed by PET/computed tomography could be more informative than morphological response [42]. Moreover, a trend toward better survival was observed in metabolic responders. In a pilot study, Ueda et al. reported that metabolic responders on FDG PET/CT after 4 weeks of neoadjuvant letrozole was associated with a significant decrease in tumor Ki67 expression level, whereas morphological response was not [43]. FDG PET or PET/CT scans should be further investigated in larger cohorts of patients treated by NET.

Axillary imaging may be more challenging than breast imaging, particularly after NET, and its clinical value needs to be evaluated. 

### 6.2. Pathological Response

Assessment of pathological response is more precise than assessment of clinical or radiological response, although measurement of pathological tumor size may also be biased. However, pathological assessment may also estimate the percentage of residual cancer cellularity. Several methods can be used to grade pathological response; the most recent and most extensively validated method is the residual cancer burden (RCB) score [44].

Pathological complete response (pCR) is rarely reported after NET, and ranges from 1–17% depending on the treatment duration (Table 3). It must be kept in mind that pCR rates after NCT range from 6 to 9% in histological grade 1 or 2 HR +/HER2- tumors [3]. 

### 6.3. Residual Cancer Burden 

Calculation of the residual cancer burden (RCB) index is based on residual breast tumor size and cellularity, and the number and size of nodal metastases. Pathological complete response is scored as RCB = 0 and residual disease is categorized into three classes RCB-I, RCB-II, and RCB-III. Given the fairly recent use of this score, only limited data are available concerning RCB after NET. In the ABCSG-34 trial, NET-treated patients (*n* = 83) were less likely to achieve RCB 0-1 (18.1%) compared to NCT-treated patients (24.6%) [47]. The RCB score was used in the NeoPAL and CORALLEEN trials, which both randomized patients to receive either NET combined with a CDK4/6 inhibitor or NCT. In NeoPAL, pathological complete response (RCB 0-I) was recorded in 7.7% (95% CI 0.4–14.9) of patients treated with letrozole + palbociclib and 15.7% (95% CI 5.7–25.7) in the NCT arm [45]. In CORALLEEN 3, the RCB 0-1 rate was 6.1% (95% CI 1.3–16.8) for patients treated with letrozole + ribociclib and 11.8% (95% CI 4.5–27.8) for those treated with NCT [46]. These results highlight the low rate of pathological response regardless of the type of neoadjuvant therapy in the HR+/HER2- population.

Although RCB is not a dichotomous variable like pCR, evaluation of the RCB index may also have a limited prognostic value after NET. Further studies are necessary to determine the value of the RCB index after NET. However, other methods of pathological response assessment for NET have become obsolete following the development of the preoperative prognostic index (PEPI) score, described below.

In our pooled analysis of the CARMINA02 and HORGEN trials, we used the Sataloff classification to assess the pathological response rate and its prognostic value after NET [18]. Somewhat similar to the RCB score, the Sataloff classification examines the therapeutic effect of neoadjuvant therapy by measuring the reduction in tumor cell number in breast and nodes [48]. It should be stressed that the 34 patients who achieved either complete or partial pathological response (defined as TA or TB combined with NA or NB in the Sataloff classification) in our pooled study were event-free at 5 years (BJC 2020). The Sataloff classification is excessively subjective and is no longer used.

The low pathological response rate after NET, regardless of how this response is assessed, make it necessary to find different surrogate endpoints.

## 7. Prognosis

Few studies have reported long-term survival data after NET. Studies using the prognosis of NET-treated patients as primary endpoint are difficult to conduct for a number of reasons. Firstly, the number of BC patients in NET studies is relatively low compared to NCT studies. Secondly, long follow-up is required, since late relapses are frequent in estrogen receptor (ER)+ BC [49]. As indicated above, mean pCR rates are 3% and are poorly correlated with survival. Finally, administration of various adjuvant therapies, including chemotherapy or alternative endocrine therapy, may influence prognosis.

Although the tumor burden often remains high, with persistent node involvement when initially present, NET-treated patients often have a good prognosis [38,50,51,52]. We found a 5-year relapse-free survival (RFS) rate of 83.7% and an overall survival rate of 92.7% in our a pooled analysis involving 217 postmenopausal women treated with anastrozole or fulvestrant for 4 to 6 months before surgery (in a population with 75% T2, 35% of which had initial lymph node involvement, and 70% grade II tumors) [18]. Interestingly, adjuvant chemotherapy, which was administered to 21.7% of patients according to each center’s local policy, was not correlated with prognosis on multivariate analysis.

## 8. Biomarkers

### 8.1. Estrogen Receptor (ER)

ER positivity is obviously predictive of response to endocrine therapy with a positive predictive value of 50–70%. The predictive role of progesterone receptor (PgR) is controversial, with some studies showing a better response in ER+/PgR+ breast cancers than ER+/PgR- breast cancers. The Allred score is obtained by adding the intensity of staining score to the percentage of ER+ cells on immunohistochemistry [53]. The total score ranges from 0 to 8, and tumors with a score ≥ 3 are considered to be HR+. This score is predictive of the response to endocrine therapy. It was demonstrated from pathological data of patients included in the P024 and IMPACT trials that the higher the baseline ER Allred score, the better the response to endocrine therapy [54,55]. Unlike the ER Allred score, the relationship between the PgR Allred score and clinical response rates did not fit a linear model. Breast cancers with an ER Allred score of 6 to 8 are therefore very good “candidates” for neoadjuvant endocrine therapy and this score was one of the inclusion criteria in the ACOSOG Z1031 trial [14].

### 8.2. Ki67 

An alternative endpoint in NET trials is Ki67 expression score at baseline and on treatment. On-treatment Ki67 expression levels reflect the ability of endocrine therapy to suppress tumor proliferation [56,57]. In the P024 trial, the baseline Ki67 expression level was not predictive of RFS, but a lower Ki67 level was correlated with better survival [10]. A significant decrease in Ki67 after 2 to 4 weeks of neoadjuvant endocrine therapy, on a new biopsy and on the surgical specimen at 12 weeks, showed a positive prognostic value in trials with aromatase inhibitors and was shown to be correlated with long-term outcome (Table 4). In the ACOSOG Z1031, no significant difference was observed between the three aromatase inhibitors letrozole, anastrozole, and exemestane in terms of Ki67 suppression after 16 weeks of treatment, together with no significant difference in terms of response rate [14]. However, improved biological activity may not actually translate into a better tumor response rate. In the IMPACT trial, overall response rates were not significantly different between tamoxifen and anastrozole, while a more marked Ki67 reduction was observed with anastrozole [11]. In the NEWEST trial, a more marked reduction in Ki67 % at week 4 (−78.8 vs. −47.4%, *p* < 0.0001) and ER expression (−25.0 vs. −13.5%, *p* = 0.0002) were observed with 500 mg neoadjuvant fulvestrant compared with 250 mg, but with no increase in response rates [17]. Further studies must focus on survival data.

These observations led to biomarker-driven treatment strategies to identify women with a low risk for disease recurrence.

The ACOSOG Z1031 protocol was amended to include tumor Ki67 expression level after 2 to 4 weeks of AI [14]. Review of data from the IMPACT [11] and POL trials [58] led to the definition of a 10% threshold, as, above this value after 1 month of NET, the chance of a PEPI 0 tumor was less than 2% [52] and poorer RFS was observed (*p* = 0.0016). Patients with Ki67 level ≥ 10% at 2–4 weeks of AI were switched to NCT in the ACOSOG Z1031B trial [14]. The efficacy threshold of 20% pCR after NCT was not met, since only two of the 35 patients included in the trial achieved a pCR [52], showing the disappointing efficacy of NCT in patients with “primary resistance” to NET. The phase III POETIC trial randomized 2:1 postmenopausal ER+ BC patients to receive 2 weeks preoperative and postoperative AI vs. no perioperative treatment [59]. The primary endpoint was time to recurrence (TTR) and the secondary endpoint was Ki67 at baseline and after 2 weeks of AI as predictor of outcome. TTR did not differ between the two groups. Patients with a low baseline Ki67 (predefined as < 10%) had a good prognosis with a 5-year recurrence rate of 4.3%. Patients with baseline Ki67 ≥ 10% who achieved a Ki67 decrease at 2 weeks also had a good prognosis with a 5-year recurrence rate of 8.4%. Patients with a high Ki67 at 2 weeks should be considered for additional chemotherapy and/or new agents, since the 5-year recurrence rate in these patients is 21.5%. 

The optimal timeframe to measure Ki67 expression level during or after endocrine therapy has not been clearly determined, probably because of the very heterogeneous patterns of decline of Ki67 levels.

Finally, it is interesting to note that the correlation between the decrease in Ki67 and improved RFS rates in the P024 [10] and IMPACT [11] trials perfectly reflects the results of their counterpart adjuvant BIG 1-98 [60] and ATAC trials [61]. Conversely, the absence of difference in terms of the decrease in Ki67 between letrozole and exemestane in trial ACOSOG Z1031 could also have predicted the negative results of trial MA27 [62]. 

### 8.3. Genomic Signatures

As in the adjuvant setting for HR+/HER2- BC, genomic signatures could be used to identify patients in whom CT can be avoided. 

Several genomic signatures (GS) have been developed for the management of ER+/HER2- early breast cancer, such as Oncotype RS, PAM50 (ROR score), Endopredict and Mammaprint [63]. Development of these GS was primarily designed to 1) assess distant recurrence and prognosis and 2) identify patients who could benefit from adjuvant chemotherapy. A high genomic signature score has been shown to be correlated with poorer prognosis and decreased benefit of adjuvant chemotherapy [64,65], and a higher pCR rate in patients treated by NCT [66,67,68]. A high GS score with these signatures is often associated with high tumor proliferation and higher grade—factors that are typically predictive of the efficacy of chemotherapy, but not necessarily predictive of the efficacy of endocrine therapy, as previously discussed. 

Nevertheless, several studies have demonstrated that a low GS score was associated with a better tumor response using NET [69,70,71,72] (Table 5). Using OncotypeDx, the TransNEOS trial (letrozole, 295 patients) found a clinical response rate of 54% when the Recurrence Score (RS) was < 18, 42% for a RS between 18–30, and 22% when RS was ≥ 31 [71]. In the study by Ueno et al., using exemestane in 64 patients, a clinical response rate of 59% was observed in patients with a low RS vs. 20% in patients with a high RS [69]. A lower RS score was also correlated with a higher BCS rate [71,72], with, in the study by Ueno et al., 91% of BCS in the low RS group, 77% in the intermediate RS group, and 47% in the high RS group [69]. Akashi-Tanaka et al., in a small cohort of 43 patients, also suggested that a low RS predicted better RFS than for patients with intermediate or high RS (5-year RFS, 100% vs. 84% and 73%, respectively) [70]. Using Endopredict in 83 patients treated with NET, a low-risk 12-gene molecular score (MS) was also associated with a higher RCB 0-1 rate (27%) than a high-risk MS score (8%) [47].

Ueno et al. also studied the prognostic value of post-NET GS scores and found a high correlation between baseline and post-NET RS scores, as baseline and post-NET scores were both associated with disease-free survival (DFS) in a cohort of 59 patients [73]. Interestingly, in this study, the combined (baseline + post-NET) RS was superior to the PEPI score and baseline or post-NET RS to predict DFS.

As the gold standard for assessment of response to NET remains debated, recent studies have used GS to evaluate the efficacy of NET. For example, in the CORALLEEN trial (neoadjuvant ribociclib + letrozole), described above, the primary endpoint to evaluate the efficacy of treatment was the proportion of patients with PAM50 low-risk-of-relapse (ROR) disease at surgery [46]. In summary, a low GS before NET is predictive of a better clinical response, BCS, RCB, and RFS.

Ki67 expression level assessed at 2–4 weeks of NET and genomic signatures are probably the most evaluated surrogate endpoints, and therefore the most often used in NET trials.

### 8.4. PIK3CA Mutations

PIK3CA encodes for the p110-α subunit of the phosphatidylinositol 3-kinase enzyme complex. PIK3CA mutations are observed in 30–40% of HR+/HER- primary and metastatic breast cancers. In the advanced setting, these mutations confer poorer prognosis and endocrine resistance [74,75].

Using 235 samples from the P024, RAD222, POL, and ACOSOG Z1031 trials, PIK3CA mutation was weakly associated with poorer clinical response and this association was not considered to be clinically meaningful [76]. In our phase II CARMINA 02 trial of neoadjuvant anastrozole or fulvestrant, PIK3CA was significantly more frequently mutated in radiological non-responders than in responders (60.8 vs. 31.6%) [19]. However, PIK3CA mutation status does not predict change in Ki67 after 2 weeks of aromatase inhibitor therapy [76,77]. Because studies yielded conflicting results in terms of endocrine responsiveness, further research on the role of PIK3CA mutations and response to NET is needed. 

### 8.5. ESR1 Mutations

The *ESR1* gene encodes the ER α unit. *ESR1* mutations have been found in 10–40% of advanced breast cancers. In this context, they confer poor prognosis and endocrine resistance, particularly to aromatase inhibitors [78]. In contrast with the advanced setting, *ESR1* mutations are detected in 0–3% of primary BC [79]. In the CARMINA02 trial, the frequency of baseline *ESR1* mutation was too low (3.4%) to draw any conclusion regarding endocrine responsiveness [19]. In a prospective cohort of 100 stage II–III HR+/HER2- BC patients treated with 3 months of anastrozole before surgery, Reinert et al. did not find any *ESR1* mutation in the 23 surgical samples of endocrine-resistant tumors, defined as having a PEPI score ≥ 4- [80]. The authors concluded that *ESR1* mutations do not appear rapidly during ET and should be considered as a mechanism of ET resistance only in advanced BC.

### 8.6. PEPI Score

Based on patients included in the P024 trial, Ellis et al. developed a preoperative prognostic index (PEPI score) validated in an independent cohort of patients from the IMPACT study [11]. This index combines the post-treatment Ki67 level with ER status, pathological tumor size, and node status (Table 6). The total PEPI score is the sum of the risk points derived from pT stage, pN stage, Ki67 level, and estrogen receptor (ER) status at surgery [50]. 

In this score, a hazard ratio (HR) of 1 or 2 for a risk factor confers one risk point, an HR of 2 to 2.5 confers two risk points, and an HR > 2.5 confers three risk points.

The total risk point score is the sum of all risk points obtained for the four factors. Based on the patients in the P024 and after validation in the IMPACT trial, the model produced a statistically significant separation of the three PEPI risk groups (low risk score = 0, intermediate risk score = 1–3, and high-risk score ≥ 4).

The PEPI score was further validated in the ACOSOG Z1031A trial [80]. The hazard of breast cancer recurrence for PEPI = 0 cases relative to the PEPI > 0 cases was 0.27 (*p* = 0.014; 95% CI, 0.092 to 0.764) when stratifying by cohort and known adjuvant chemotherapy use. After a median follow-up of 5.5 years, 3.7% of the patients with a PEPI score of 0 experienced relapse versus 14.4% of patients with PEPI > 0 (HR = 0.27; 95% CI: 0.092–0.764; *p* = 0.014), supporting the use of ET only in PEPI 0. Longer-term outcome data from this trial are awaited.

We also evaluated prognosis based on the PEPI score in 217 patients treated with anastrozole or fulvestrant in the pooled analysis of CARMINA02 and HORGEN trials [18]. In this pooled analysis, PEPI group III (score ≥ 4) compared with PEPI groups I or II was the only variable significantly associated with poorer 5-year RFS on multivariate analysis. Unlike the results of the ACOSOG Z1031 trial [81], no statistically significant survival differences were found between patients with PEPI score 0 and those with PEPI score > 0, although a trend was observed (*p* = 0.06).

It is important to note that a modified PEPI score, including pathological tumor size, node status, and post-treatment Ki67 level without ER status, has been proposed for patients receiving fulvestrant, as this drug downregulates ER. mPEPI appears to have a similar prognostic value to that of PEPI [81]. 

The previously mentioned ALTERNATE trial is a biomarker-driven treatment strategy to identify women with a low risk for disease recurrence. Neoadjuvant anastrozole and/or fulvestrant are given for 24 weeks except for patients with a Ki67 ≥ 10% at 4 or 12 weeks, who then receive NCT. Co-primary endpoints are the rate of endocrine-sensitive disease (mPEPI = 0 or pCR) and 5-year RFS. Adjuvant CT is only administered to the mPEPI > 0 group. Preliminary results have shown no difference in the ESDR between the three treatment arms: 18.6%, 21.8%, and 20% in the anastrozole, fulvestrant, and combination arms, respectively [21]. Such biomarkers driven strategies illustrate the use of NET as a prognostic tool in ER+/HER2- BC. 

In Figure 1 a proposal for optimal NET management is summarized.

## 9. Targeted Therapies

Several therapies that target potential endocrine resistance pathways have been associated with NET. In particular, endocrine therapy could be combined with PIK3CA inhibitors, since activation of the PIK3CA pathway, particularly via PIK3CA gene mutations, is a mechanism of resistance to ET. In a randomized phase II trial, 270 postmenopausal BC patients received letrozole plus the mTOR inhibitor everolimus vs. letrozole + placebo for 4 months before surgery [82]. The primary endpoint was clinical response rate by palpation, which was significantly higher in the everolimus group (68.1% vs. 59.1%; one-sided χ^2^ test *p* = 0.0616; limit of significance *p* < 0.10). The large-scale phase II LORELEI trial including 334 stage I–III postmenopausal women demonstrated that adding taselisib (an α-selective PIK3CA inhibitor) to letrozole resulted in a higher response rate assessed by MRI of 50% vs. 39% in the placebo group. Low rates of pathological response, which was the co-primary endpoint, were observed with no difference between the two arms: 2% with taselisib vs. 1% with placebo [83]. Alpelisib is another α-selective PIK3CA inhibitor that demonstrated a significant improvement in progression-free survival when added to fulvestrant for HR+/HER2- PIK3CA-mutant advanced BC in the SOLAR-1 trial [84]. In contrast, addition of alpelisib to 24 weeks of neoadjuvant letrozole did not improve clinical or pathological response rates regardless of PIK3CA mutation status [85]. Several other targeted treatments, such as gefitinib or lapatinib, have been combined with NET. However, most of the results were disappointing in terms of response, with increased toxicity in the targeted treatment arm [13]. 

Given the efficacy of CDK4/6 inhibitors in the metastatic setting, NET could be combined with these targeted therapies. In the NeoPALAna study, 50 patients with stage II-III ER+/HER2- BC received anastrozole for 4 weeks followed by anastrozole + palbociclib for 4 months. The primary endpoint was complete cell cycle arrest, i.e., a Ki67 expression level < 2.7% [86]. Adding palbociclib led to higher cycle arrest rates. Several phase II trials of neoadjuvant therapy combining ET + CDK4/6 inhibitors have subsequently been conducted. In the PALLET trial, which included 370 patients, palbociclib added to letrozole vs. letrozole also enhanced the reduction of Ki67 levels (median log-fold change −4.1 vs. −2.2; *p* = 0.001) in 190 evaluable patients. However, clinical response rates were similar, whether or not patients received palbociclib (54.3% v 49.5%; *p* = 0.02) [87]. 

The results of the neoMONARCH study, which combined abemaciclib with anastrozole, showed a significant reduction in Ki67 levels with CDK4/6 inhibitor with or without anastrozole, compared to anastrozole alone after 14 weeks of therapy [88]. The phase II FELINE trial was designed to determine whether neoadjuvant therapy with ribociclib combined with letrozole for 24 weeks increases the proportion of patients with a PEPI score of 0 at surgery, compared with single agent letrozole. Preliminary results did not show any difference between the two treatment arms [89].

The NeoPAL trial randomly assigned 106 patients with luminal A or B breast cancer as determined by PAM50 to 19 weeks of palbociclib and letrozole or NCT (three cycles of FEC followed by three cycles of docetaxel). The primary objective was the proportion of patients with an RCB index of 0–1 [45]. Only 7.7% and 15.7% of patients achieved RCB 0–1 in the two arms, respectively. Clinical response and BCS rates were similar. Moreover, the toxicity profile was in favor of the letrozole–palbociclib arm.

In the CORALLEEN trial, 106 postmenopausal women with Luminal B breast cancer as determined by PAM50 were randomized to receive six cycles of letrozole + ribociclib or four cycles of AC followed by weekly paclitaxel for 12 weeks [46]. PAM50 low-ROR at surgery was used as primary endpoint: 46.9% of tumors in the letrozole-ribociclib arm and 46.1% in the chemotherapy arm achieved a low ROR at surgery. A pCR was observed in three patients in the CT arm and no patients in the letrozole + ribociclib arm, and a PEPI score of 0 was observed in 17.3% and 22.4% of patients, respectively. BCS rates were comparable between letrozole + ribociclib (85.7%) and NCT (72.2%). Once again, chemotherapy was more toxic, particularly in terms of febrile neutropenia.

The findings of the CORALLEEN and NeoPAL trials are not sufficiently conclusive to adopt this strategy as a standard of care. However, they are encouraging and could help to identify patients in whom chemotherapy is unnecessary. Survival outcomes of these trials are awaited. 

## 10. Candidates for NET

Although the use of NET increased slightly between 2004 and 2012, this modality was rarely used (3% of patients) among the 77,272 patients with T2-T4c HR+ BC included in the National Cancer Data Base [90]. Analysis of the patient population included in NET trials nevertheless suggests that postmenopausal women with highly ER-positive (Allred score 6-8)/HER2-negative breast cancer that is too large for BCS or inoperable at diagnosis, are very good candidates for NET. NET might even be more appropriate in this population than NCT, bearing in mind that comparative studies are rare and that meta-analyses of adjuvant CT have shown that chemotherapy improves outcome in ER+ BC, although not exactly to the same extent as in ER-negative disease [91].

An unresolved issue is the value of the baseline tumor proliferation rate and/or genomic signatures as criteria for choosing between NET and NCT. The results of several of the abovementioned studies do not detract from the role of NET +/− targeted therapies in highly proliferative BC, provided a decrease in the proliferative index is observed after 2 to 4 weeks of treatment, as in the POETIC trial.

## 11. Conclusions

Over the last ten years, NET has been increasingly investigated and has gained recognition. NET should not only be used to allow surgery or to improve BCS rates in patients not eligible for NCT, but also as a research tool for the search for endocrine sensitivity biomarkers and targeted therapies, and for prognostic information in ER+/HER2- BC. Indeed, one of the major ongoing issues in this subtype is to determine in which patients chemotherapy can be safely spared.

## Figures and Tables

**Figure 1 cancers-13-00902-f001:**
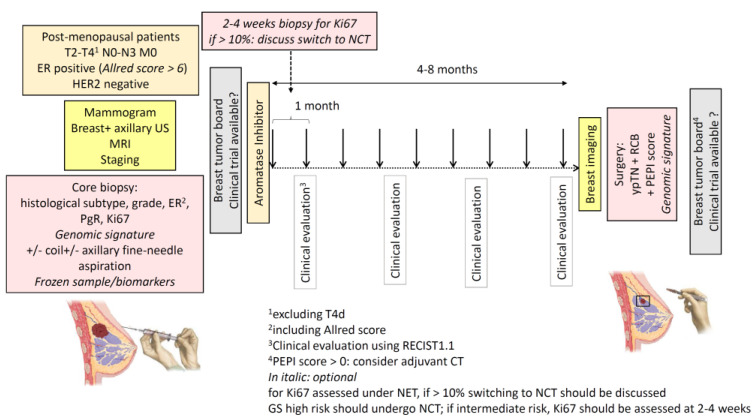
Proposal for optimal NET management. Postmenopausal women with clinical stage II or III ER‑positive, HER2-negative breast cancer are eligible for NET. Initial evaluation should include: mammogram, breast and axillary US (+/− concomitant fine-needle aspiration for nodal status), breast MRI and staging; a core biopsy is needed for assessment of histological subtype, histological grade, hormone receptors, Ki67 and optional genomic signature. After initial tumor board, an aromatase inhibitor should be given if no clinical trial is available. Clinical evaluations at 1 month and then every 3 to 4 months should be performed. Tumor biopsy 2 to 4 weeks after NET initiation may be done for Ki67 assessment and a switch for NCT encouraged if Ki67 > 10%. After 4 to 8 months of NET, surgery -after breast imaging- should be planned if feasible. Pathological examination should collect ypTN, RCB, PEPI score and optional genomic signature. Adjuvant treatments are decided according to the local policy if no clinical trial is available. Adjuvant chemotherapy is not recommended for patients with PEPI score 0. Abbreviations: ER: estrogen receptor; PgR: progesterone receptor; MRI: magnetic resonance imaging; NCT: neoadjuvant chemotherapy; RCB: residual cancer burden; PEPI: preoperative endocrine prognostic index; GS: genomic signature.

**Table 1 cancers-13-00902-t001:** Randomized trials comparing aromatase inhibitor (AI) vs. tamoxifen or AI vs. AI. * Significant (*p* < 0.05).

		*n*	Treatment	CR	pCR	BCS
IMPACT [11]	2005	330	A 12 weeks *vs.* T 12 weeks *vs.* T + A 12 weeks → surgery	37% 36% 39%		46% 31% 24%
PROACT [12]	2006	451	A 3 months *vs.* T 3 months → surgery	39.5% 35.4%		43% * 31%
P024 [10]	2001	337	L 4 months *vs.* T 4 months → surgery	55% * 36%	1.3% 1.8%	45% * 35%
ACOSOG Z1031 [14]	2011	374	L 16–18 weeks *vs.* A 16–18 weeks *vs.* E 16–18 weeks → surgery	74% 69% 62%		41% 64% 48%
STAGE [22]	2012	197	A + G 24 weeks *vs.* T + G 24 weeks → surgery	70.4% * 50.5%	1% 0	86% 68%

A = anastrozole; T = tamoxifen; L = letrozole; E = exemestane; G = goserelin; CR = clinical response; pCR = pathological complete response; BCS = breast-conserving surgery.

**Table 2 cancers-13-00902-t002:** Randomized trials comparing neoadjuvant chemotherapy (NCT) vs. neoadjuvant endocrine therapy (NET).

		*n*	Treatment	CR	pCR	BCS
Semiglazov [24]	2007	239	Doxo-Pacli *vs.* A or E 3 months → surgery	64% 64%	6% 3%	24% 33%
GEICAM 2006–03 [23]	2012	95	EC-Doce *vs.* E → surgery	66% 48%	<1% 0	47% 56%
NEOCENT [26]	2014	44	chemo *vs.* L 3–4 months →surgery	54% 59%	0 0	55% 68%
Kim [25]	2020	187	AC-Pacli *vs.* T + G 24 weeks →surgery	84% 53%	3.4% 1.2%	55% 46%

Doxo = doxorubicin; Pacli = paclitaxel; A = anastrozole; E = exemestane; EC = epirubicin cyclophosphamide; Doce = docetaxel; L = letrozole; AC = doxorubicin cyclophosphamide; T = tamoxifen; G = goserelin; CR = clinical response; pCR = pathological complete response; BCS = breast-conserving surgery.

**Table 3 cancers-13-00902-t003:** Pathological complete response (pCR) rates after 3 to 12 months of NET.

NET	pCR %
P024 [10]	1.5
Allevi [28]	
4 months	2.5
8 months	5
12 months	17.5
Semiglazov 2007 [24]	3
GEICAM 2006-03 [23]	0
NEOCENT [26]	0
ACOSOG Z1031 [14]	1.6
NeoPAL ^1^ [45]	3.8
CORALLEEN ^2^ [46]	0

^1,2^ Patients included in NeoPAL ^1^ and CORALLEEN trials ^2^ who were randomized to receive ET also received palbociclib or ribociclib, respectively.

**Table 4 cancers-13-00902-t004:** Comparison of reductions in Ki67 expression levels with different NET at various time-points and correlation with RFS.

Ki67	*n*	2 Weeks	12 Weeks	RFS
P024 [10] LET vs. TAM	337		87 vs. 75% *p* = 0.0009	*p* < 0.001
IMPACT [11] ANA vs. TAM	330	76 vs. 60% *p* = 0.004	82 vs. 62% *p* < 0.001	*p* = 0.004
ACOSOG Z1031 [14] LET vs. ANA vs. EXE	377		87 vs. 81 vs. 78%	NS

**Table 5 cancers-13-00902-t005:** Neoadjuvant endocrine therapy efficacy according to genomic score.

Study	*N* =	Clinical Response Rate	Breast-Conserving Surgery Rate	Pathological Response	Relapse-Free Survival
**OncotypeDx**					
TransNEOS trial (71)	295	Low RS: 54% Inter RS: 42% High RS: 22%	Low RS: 79% High RS: 60%	NR	NR
JFMC34-0601 Ueno et al. (69)(73)	64 and 59	Low RS: 59% High RS: 20%	Low RS: 91% Inter RS: 77% High RS: 47%	NR	Low RS: 90% Inter RS: 75% High RS: 50% Combined RS (pre/post NET): no recurrence in the low RS group.
Akashi-Tanaka et al. (70)	43	Low RS: 64% Inter RS: 31% High RS: 31%	NR	NR	Low RS: 100% Inter RS: 84% High RS: 73%
Bear (72) Low RS < 11 Inter RS 11-25	30	Low RS 83% Inter RS: 50%	Low RS 75% Inter RS: 72%	pCR Low RS: 8.3% Inter RS: 6%	NR
**Endopredict**					
ABCSG-34 trial(47)	83	NR	NR	RCB0-I Low MS 27% High MS 8%	NR

NR: not reported; Inter: intermediate; pCR: pathological complete response; MS: 12-gene molecular score. If not specified, low RS < 18 and high RS ≥ 31.

**Table 6 cancers-13-00902-t006:** Preoperative endocrine prognostic index.

Post-NET Pathology and Biomarker Status	Relapse-Free Survival		Breast Cancer-Free Survival	
	HR	Points	HR	Points
Tumor size				
pT1 or pT2	-	0	-	0
pT3 or pT4	2.8	3	4.4	3
Node status				
Negative (pN0)	-	0	-	0
Positive (pN-3)	3.2	3	3.9	3
Ki67 level				
0–2.7%	-	0	-	0
>2.7–7.3%	1.3	1	1.4	1
>7.3–19.7%	1.7	1	2	2
19.7–53.1%	2.2	2	2.7	3
>53.1%	2.9	3	3.8	3
ER, Allred score				
0–2	2.8	3	7	3
3–8	-	0	-	0

Abbreviations: ER, estrogen receptor; HR, hazard ratio.

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
