# Peer review of "Neoadjuvant Endocrine Therapy in Breast Cancer Management: State of the Art"

_cancers, 2021, doi:10.3390/cancers13040902_

Round 1

Reviewer 1 Report

The work presented by Lerebours et al. provides an extensive and accurate analysis of progress in neoadjuvant therapy for BC ER+ toumors, ponting out main issues and challenges.

The work is complete and well described.

I suggest some stylistic improvement:

  • In paragraph “Duration of net” page 4, it seems that a reference in last 5 lines is missing. I suggest to add it
  • In paragraph “biomarkers, ER” page 7, the sentence “Based on characteristics of patients ….” looks confusing. I suggest of rewrite it.
  • In paragraph “Genomic signatures” page 8, line 5 I suggest to add Endopredict to the mentioned GS. I also suggest to insert a table with main GS differences and relative clinical trials, and call back the table in the text.
  • Why do you have inserted the paragraph prognosis as subgroup of biomarkers? i suggested to insert it as a separate paragraph, as well the paragraph targeted therapies.

Author Response

Reviewer 1

The work presented by Lerebours et al. provides an extensive and accurate analysis of progress in neoadjuvant therapy for BC ER+ tumors, pointing out main issues and challenges.

The work is complete and well described.

I suggest some stylistic improvement:

  • In paragraph “Duration of net” page 4, it seems that a reference in last 5 lines is missing. I suggest to add it

We thank the reviewer for this comment and have added a reference  (Macaskill et al, 2007), now reference 30.

  • In paragraph “biomarkers, ER” page 7, the sentence “Based on characteristics of patients ….” looks confusing. I suggest of rewrite it.

We agree with the reviewer’s comment and have changed this sentence as follows: It was demonstrated from pathological data of patients included in the P024 and IMPACT trials that the higher the baseline ER Allred score, the better the response to endocrine therapy.

  • In paragraph “Genomic signatures” page 8, line 5 I suggest to add Endopredict to the mentioned GS. I also suggest to insert a table with main GS differences and relative clinical trials, and call back the table in the text.

We completely agree with the reviewer’s comment and apologize for omitting Endopredict in the list of genomic signatures. According to this comment, we have also included a table on main GS differences and relative trials (table 5).

  • Why do you have inserted the paragraph prognosis as subgroup of biomarkers? I suggested to insert it as a separate paragraph, as well the paragraph targeted therapies.

We thank the reviewer for this suggestion. We have now moved the paragraph “Prognosis” before the paragraph on biomarkers. The last paragraph is on targeted therapies. In addition, the paragraphs have now been numbered to make the reading easier. “Prognosis” is now paragraph 6 and “Biomarkers” paragraph 7.

Reviewer 2 Report

That is a quite good review regarding endocrine therapy in breast cancer treatment. Since that is a nice review of patient trails that where already published there is not much to add.

However, some points need to be addressed in order for this review would be beneficial:

Major:

  • The title of the review is not full. To my understanding authors intention was to state “Neoadjuvant Endocrine Therapy in Breast Cancer treatment: state of the art” or at least something like that. Then, terming “state of art” instead of “cutting edge” or “up to date” is already a preference of author’s style.
  • There is a huge lack of visual representation. At least a graphical abstract or some other sort of visualization is preferred.
  • The text mainly composes of the mentioning of clinical trials and discussing just numbers. From one side it is ok. However, as I understand, authors wanted to summarise that the Neoadjuvant Endocrine Therapy has more potential as compared to today’s conventional usage. I drow such conclusion from first sentence of conclusions: “NET should not only be used to allow surgery or to improve BCS rates in patients not eligible for NCT, but also as a research tool”. The delivery of this message has failed. Authors must “extract” the main message in more clear way. As a suggestion it might by some scheme or a graph or rearranging text itself. So, to summarise, the gathered data presentation must be improved. Nevertheless, other message from the statement “The purpose of this review is to discuss the practical modalities of NET, current research on predictive factors and new drugs, and finally to define the population to whom NET should be offered” was delivered in understandable manner.
  • Could authors explain why Neoadjuvant Endocrine Therapy should be “also as a research tool”? Up to now it looks like it’s word without a message, because everything can be used as a research tool: hammer, CT scan or clinical trial itself.

Author Response

Reviewer 2

  • The title of the review is not full. To my understanding authors intention was to state “Neoadjuvant Endocrine Therapy in Breast Cancer treatment: state of the art” or at least something like that. Then, terming “state of art” instead of “cutting edge” or “up to date” is already a preference of author’s style.

We thank the reviewer for the suggestion on completing the title. If we understood this comment correctly, we should add the term ‘treatment’. As this word could be redundant with “endocrine therapy”, we suggest using the word “management”. We then propose to change the title of the manuscript as following: ‘Neoadjuvant Endocrine Therapy in Breast Cancer management: state of the art’.

  • There is a huge lack of visual representation. At least a graphical abstract or some other sort of visualization is preferred.

We thank the reviewer for this comment. We fully agree and have added a figure with a graphic including the main steps required when NET is chosen as treatment (now figure 1).

  • The text mainly composes of the mentioning of clinical trials and discussing just numbers. From one side it is ok. However, as I understand, authors wanted to summarise that the Neoadjuvant Endocrine Therapy has more potential as compared to today’s conventional usage. I drow such conclusion from first sentence of conclusions: “NET should not only be used to allow surgery or to improve BCS rates in patients not eligible for NCT, but also as a research tool”. The delivery of this message has failed. Authors must “extract” the main message in more clear way. As a suggestion it might by some scheme or a graph or rearranging text itself. So, to summarise, the gathered data presentation must be improved. Nevertheless, other message from the statement “The purpose of this review is to discuss the practical modalities of NET, current research on predictive factors and new drugs, and finally to define the population to whom NET should be offered” was delivered in understandable manner.

We thank the reviewer for this relevant comment. Indeed, we would like to discuss that, beyond routine care, neoadjuvant treatments are also a formidable opportunity to investigate new drugs, design adjuvant trials, and evaluate predictive biomarkers. In particular, in HR+/HER2- BC, NET offers the best way to verify the in vivo endocrine responsiveness. We have added several sentences highlighted in red at the end of paragraphs 2, 5 and in paragraph 7, about Ki67 and genomic signatures to emphasize the value of NET as a research tool.

  • Could authors explain why Neoadjuvant Endocrine Therapy should be “also as a research tool”? Up to now it looks like it’s word without a message, because everything can be used as a research tool: hammer, CT scan or clinical trial itself.

We agree with the reviewer that this wording is incomplete. We have modified the conclusion and summarized the main reasons why NET is also a research tool:

“Over the last ten years, NET has been increasingly investigated and has gained recognition. NET should not only be used to allow surgery or to improve BCS rates in patients not eligible for NCT, but also as a research tool for the search for endocrine sensitivity biomarkers and targeted therapies, but also for prognostic information in ER+/HER2- BC. Indeed, one of the major ongoing issues in this subtype is to determine in which patients, chemotherapy can be safely spared.”

Round 2

Reviewer 2 Report

Many of my concerns had been adressed. Therefore, the manuscript turned in to a publishable material.